# GENERALIZATION IN DIFFUSION MODELS ARISES FROM GEOMETRY-ADAPTIVE HARMONIC REPRESENTATIONS

**Zahra Kadkhodaie**
Ctr. for Data Science, New York University
zk388@nyu.edu

**Florentin Guth**
Ctr. for Data Science, New York University
Flatiron Institute, Simons Foundation
florentin.guth@nyu.edu

**Eero P. Simoncelli**
New York University
Flatiron Institute, Simons Foundation
eero.simoncelli@nyu.edu

**Stéphane Mallat**
Collège de France
Flatiron Institute, Simons Foundation
stephane.mallat@ens.fr

## ABSTRACT

Deep neural networks (DNNs) trained for image denoising are able to generate high-quality samples with score-based reverse diffusion algorithms. These impressive capabilities seem to imply an escape from the curse of dimensionality, but recent reports of memorization of the training set raise the question of whether these networks are learning the "true" continuous density of the data. Here, we show that two DNNs trained on non-overlapping subsets of a dataset learn nearly the same score function, and thus the same density, when the number of training images is large enough. In this regime of strong generalization, diffusion-generated images are distinct from the training set, and are of high visual quality, suggesting that the inductive biases of the DNNs are well-aligned with the data density. We analyze the learned denoising functions and show that the inductive biases give rise to a shrinkage operation in a basis adapted to the underlying image. Examination of these bases reveals oscillating harmonic structures along contours and in homogeneous regions. We demonstrate that trained denoisers are inductively biased towards these geometry-adaptive harmonic bases since they arise not only when the network is trained on photographic images, but also when it is trained on image classes supported on low-dimensional manifolds for which the harmonic basis is suboptimal. Finally, we show that when trained on regular image classes for which the optimal basis is known to be geometry-adaptive and harmonic, the denoising performance of the networks is near-optimal.

## 1 INTRODUCTION

Deep neural networks (DNNs) have demonstrated ever-more impressive capabilities for sampling from high-dimensional image densities, most recently through the development of diffusion methods. These methods operate by training a denoiser, which provides an estimate of the score (the gradient of the log of the noisy image distribution). The score is then used to sample from the corresponding estimated density, using an iterative reverse diffusion procedure (Sohl-Dickstein et al., 2015; Song & Ermon, 2019; Ho et al., 2020; Kadkhodaie & Simoncelli, 2020). However, approximating a continuous density in a high-dimensional space is notoriously difficult: do these networks actually achieve this feat, learning from a relatively small training set to generate high-quality samples, in apparent defiance of the curse of dimensionality? If so, this must be due to their inductive biases, that is, the restrictions that the architecture and optimization place on the learned denoising function. But the approximation class associated with these models is not well understood. Here, we take several steps toward elucidating this mystery.

Several recently reported results show that, when the training set is small relative to the network capacity, diffusion generative models do not approximate a continuous density, but rather memorize samples of the training set, which are then reproduced (or recombined) when generating new samples

---

Source code: https://github.com/LabForComputationalVision/memorization_generalization_in_diffusion_models

(Somepalli et al., 2023; Carlini et al., 2023). This is a form of overfitting (high model variance). Here, we confirm this behavior for DNNs trained on small data sets, but demonstrate that these same models do not memorize when trained on sufficiently large sets. Specifically, we show that two denoisers trained on sufficiently large non-overlapping sets converge to essentially the same denoising function. That is, the learned model becomes independent of the training set (i.e., model variance falls to zero). As a result, when used for image generation, these networks produce nearly identical samples. These results provide stronger and more direct evidence of generalization than standard comparisons of average performance on train and test sets. This generalization can be achieved with large but realizable training sets (for our examples, roughly $10^5$ images suffices), reflecting powerful inductive biases of these networks. Moreover, sampling from these models produces images of high visual quality, implying that these inductive biases are well-matched to the underlying distribution of photographic images (Wilson & Izmailov, 2020; Goyal & Bengio, 2022; Griffiths et al., 2023).

To study these inductive biases, we develop and exploit the relationship between denoising and density estimation. We find that DNN denoisers trained on photographic images perform a shrinkage operation in an orthonormal basis consisting of harmonic functions that are adapted to the geometry of features in the underlying image. We refer to these as *geometry-adaptive harmonic bases* (GAHBs). This observation, taken together with the generalization performance of DNN denoisers, suggests that optimal bases for denoising photographic images are GAHBs and, moreover, that inductive biases of DNN denoisers encourage such bases. To test this more directly, we examine a particular class of images whose intensity variations are regular over regions separated by regular contours. A particular type of GAHB, known as "bandlets" (Peyré & Mallat, 2008), have been shown to be near-optimal for denoising these images (Dossal et al., 2011). We observe that the DNN denoiser operates within a GAHB similar to a bandlet basis, also achieving near-optimal performance. Thus the inductive bias enables the network to appropriately estimate the score in these cases.

If DNN denoisers induce biases towards the GAHB approximation class, then they should perform sub-optimally for distributions whose optimal bases are not GAHBs. To investigate this, we train DNN denoisers on image classes supported on low-dimensional manifolds, for which the optimal denoising basis is only partially constrained. Specifically, an optimal denoiser (for small noise) should project a noisy image on the tangent space of the manifold. We observe that the DNN denoiser closely approximates this projection, but also partially retains content lying within a subspace spanned by a set of additional GAHB vectors. These suboptimal components reflect the GAHB inductive bias.

## 2 DIFFUSION MODEL VARIANCE AND DENOISING GENERALIZATION

Consider an unknown image probability density, $p(x)$. Rather than approximating this density directly, diffusion models learn the scores of the distributions of noise-corrupted images. Here, we show that the denoising error provides a bound on the density modeling error, and use this to analyze the convergence of the density model.

### 2.1 DIFFUSION MODELS AND DENOISING

Let $y = x + z$ where $z \sim \mathcal{N}(0, \sigma^2 \mathrm{Id})$. The density $p_\sigma(y)$ of noisy images is then related to $p(x)$ through marginalization over $x$:

$$p_\sigma(y) = \int p(y|x)\, p(x)\, \mathrm{d}x = \int g_\sigma(y - x)\, p(x)\, \mathrm{d}x, \tag{1}$$

where $g_\sigma(z)$ is the density of $z$. Hence, $p_\sigma(y)$ is obtained by convolving $p(x)$ with a Gaussian with standard deviation $\sigma$. The family of densities $\{p_\sigma(y); \sigma \geq 0\}$ forms a scale-space representation of $p(x)$, analogous to the temporal evolution of a diffusion process.

Diffusion models learn an approximation $s_\theta(y)$ (dropping the $\sigma$ dependence for simplicity) of the scores $\nabla \log p_\sigma(y)$ of the blurred densities $p_\sigma(y)$ at all noise levels $\sigma$. The collection of these score models implicitly defines a model $p_\theta(x)$ of the density of clean images $p(x)$ through a reverse diffusion process. The error of the generative model, as measured by the KL divergence between $p(x)$ and $p_\theta(x)$, is then controlled by the integrated score error across all noise levels (Song et al., 2021):

$$D_{\mathrm{KL}}(p(x) \,\|\, p_\theta(x)) \leq \int_0^\infty \mathbb{E}_y \Big[ \|\nabla \log p_\sigma(y) - s_\theta(y)\|^2 \Big] \sigma \, \mathrm{d}\sigma. \tag{2}$$

The key to learning the scores is an equation due to Robbins (1956) and Miyasawa (1961) (proved in Appendix D.1 for completeness) that relates them to the mean of the corresponding posteriors:

$$\nabla \log p_\sigma(y) = (\mathbb{E}_x[x \mid y] - y)/\sigma^2. \tag{3}$$

The score is learned by training a denoiser $f_\theta(y)$ to minimize the mean squared error (MSE) (Raphan & Simoncelli, 2011; Vincent, 2011):

$$\text{MSE}(f_\theta, \sigma^2) = \mathbb{E}_{x,y}\Big[\|x - f_\theta(y)\|^2\Big], \tag{4}$$

so that $f_\theta(y) \approx \mathbb{E}_x[x \mid y]$. This estimated conditional mean is used to recover the estimated score using eq. (3): $s_\theta(y) = (f_\theta(y) - y)/\sigma^2$. As we show in Appendix D.2, the error in estimating the density $p(x)$ is bounded by the integrated optimality gap of the denoiser across noise levels:

$$D_{\text{KL}}(p(x) \,\|\, p_\theta(x)) \leq \int_0^\infty \Big(\text{MSE}(f_\theta, \sigma^2) - \text{MSE}(f^\star, \sigma^2)\Big)\sigma^{-3}\,\mathrm{d}\sigma, \tag{5}$$

where $f^\star(y) = \mathbb{E}_x[x \mid y]$ is the optimal denoiser. Thus, learning the true density model is equivalent to performing optimal denoising at all noise levels. Conversely, a suboptimal denoiser introduces a score approximation error, which in turn can result in an error in the modeled density.

Generally, the optimal denoising function $f^\star$ (as well as the "true" distribution, $p(x)$) is unknown for photographic images, which makes numerical evaluation of sub-optimality challenging. We can however separate deviations from optimality arising from model bias and model variance. Model variance measures the size of the approximation class, and hence the strength (or restrictiveness) of the inductive biases. It can be evaluated without knowledge of $f^\star$. Here, we define generalization as near-zero model variance (i.e., an absence of overfitting), which is agnostic to model bias. This is the subject of Section 2.2. Model bias measures the distance of the true score to the approximation class, and thus the alignment between the inductive biases and the data distribution. In the context of photographic images, visual quality of generated samples can be a qualitative indicator of the model bias, although high visual quality does not necessarily guarantee low model bias. We evaluate model bias in Section 3.2 by considering synthetic image classes for which $f^\star$ is approximately known.

## 2.2 TRANSITION FROM MEMORIZATION TO GENERALIZATION

DNNs are susceptible to overfitting, because the number of training examples is typically small relative to the model capacity. Since density estimation, in particular, suffers from the curse of dimensionality, overfitting is of more concern in the context of generative models. An overfitted denoiser performs well on training images but fails to generalize to test images, resulting in low-diversity generated images. Consistent with this, several papers have reported that diffusion models can memorize their training data (Somepalli et al., 2023; Carlini et al., 2023; Dar et al., 2023; Zhang et al., 2023). To directly assess this, we compared denoising performance on training and test data for different training set sizes $N$. We trained denoisers on subsets of the (downsampled) CelebA dataset (Liu et al., 2015) of size $N = 10^0, 10^1, 10^2, 10^3, 10^4, 10^5$. We used a UNet architecture (Ronneberger et al., 2015), which is composed of 3 convolutional encoder and decoder blocks with rectifying non-linearities. These denoisers are universal and blind: they operate on all noise levels without having noise level as an input Mohan* et al. (2020). Networks are trained to minimize mean squared error (4). See Appendix A for architecture and training details.

Results are shown in Figure 1. When $N = 1$, the denoiser essentially memorizes the single training image, leading to a high test error. Increasing $N$ substantially increases the performance on the test set while worsening performance on the training set, as the network transitions from memorization to generalization. At $N = 10^5$, empirical test and train error are matched for all noise levels.

To investigate this generalization further, we train denoisers on *non-overlapping* subsets of CelebA of various size $N$. We then generate samples using the scores learned by each denoiser, through the reverse diffusion algorithm of Kadkhodaie & Simoncelli (2020)—see Appendix A for details. Figure 2 shows samples generated by these denoisers, initialized from the same noise sample. For small $N$, the networks memorize their respective training images. However, for large $N$, the networks converge to the same score function (and thus sample from the same model density), generating nearly identical samples. This surprising behavior provides a much stronger demonstration of convergence than comparison of average train and test performance.

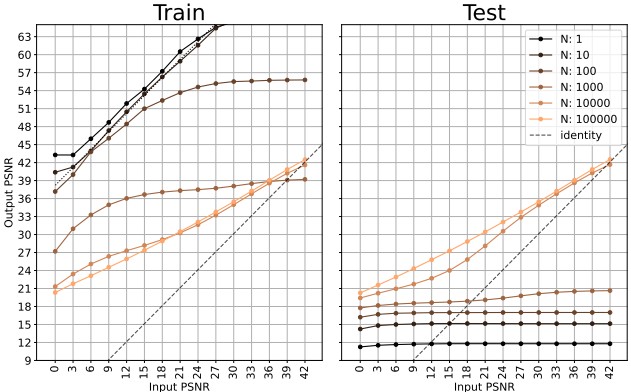

Figure 1: Transition from memorization to generalization, for a UNet denoiser trained on face images. Each curve shows the denoising error (output PSNR, ten times log10 ratio of squared dynamic range to MSE) as a function of noise level (input PSNR), for a training set of size $N$. As $N$ increases, performance on the training set generally worsens (left), while performance on the test set improves (right). For $N = 1$ and $N = 10$, the train PSNR improves with unit slope, while test PSNR is poor, independent of noise level, a sign of memorization. The increase in test performance on small noise levels at $N = 1000$ is indicative of the transition phase from memorization to generalization. At $N = 10^5$, test and train PSNR are essentially identical, and the model is no longer overfitting the training data.

## 3 INDUCTIVE BIASES

The number of samples needed for estimation of an arbitrary probability density grows exponentially with dimensionality (the "curse of dimensionality"). As a result, estimating high-dimensional distributions is only feasible if one imposes strong constraints or priors over the hypothesis space. In a diffusion model, these arise from the network architecture and the optimization algorithm, and are referred to as the inductive biases of the network (Wilson & Izmailov, 2020; Goyal & Bengio, 2022; Griffiths et al., 2023). In Section 2.2, we demonstrated that DNN denoisers can learn scores (and thus a density) from relatively small training sets. This generalization result, combined with the high quality of sampled images, is evidence that the inductive biases are well-matched to the "true" distribution of images, allowing the model to rapidly converge to a good solution through learning. On the contrary, when inductive biases are not aligned with the true distribution, the model will arrive at a poor solution with high model bias.

For diffusion methods, learning the right density model is equivalent to performing optimal denoising at all noise levels (see Section 2.1). The inductive biases on the density model thus arise directly from inductive biases in the denoiser. This connection offers a means of evaluating the accuracy of the learned probability models, which is generally difficult in high-dimensions.

### 3.1 DENOISING AS SHRINKAGE IN AN ADAPTIVE BASIS

The inductive biases of the DNN denoiser can be studied through an eigendecomposition of its Jacobian. We describe the general properties that are expected for an optimal denoiser, and examine several specific cases for which the optimal solution is partially known.

**Jacobian eigenvectors as an adaptive basis.** To analyze inductive biases, we perform a local analysis of a denoising estimator $\hat{x}(y) = f(y)$ by looking at its Jacobian $\nabla f(y)$. For simplicity, we assume that the Jacobian is symmetric and non-negative (we show below that this holds for the optimal denoiser, and it is approximately true of the network Jacobian (Mohan* et al., 2020)). We can then diagonalize it to obtain eigenvalues $(\lambda_k(y))_{1 \le k \le d}$ and eigenvectors $(e_k(y))_{1 \le k \le d}$.

If $f(y)$ is computed with a DNN denoiser with no additive "bias" parameters, its input-output mapping is piecewise linear, as opposed to piecewise affine (Mohan* et al., 2020; Romano et al., 2017). It

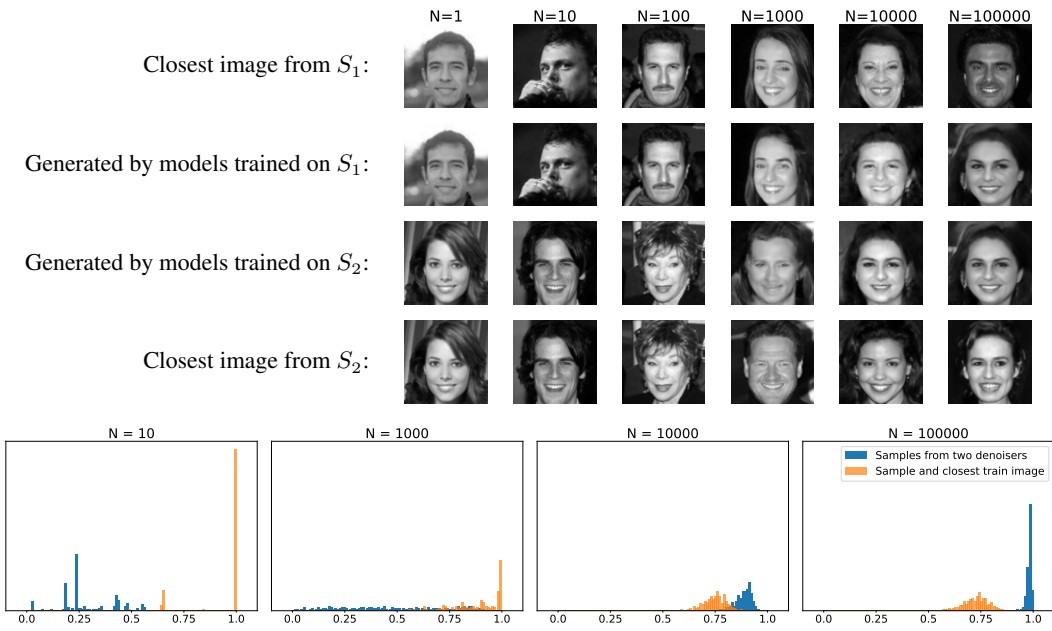

Figure 2: Convergence of model variance. Diffusion models are trained on non-overlapping subsets $S_1$ and $S_2$ of a face dataset (filtered for duplicates). The subset size $N$ varies from 1 to $10^5$. We then generate a sample from each model with a reverse diffusion algorithm, initialized from the same noise image. **Top.** For training sets of size $N = 1$ to $N = 100$, the networks memorize, producing samples nearly identical to examples from the training set. For $N = 1000$, generated samples are similar to a training example, but show distortions in some regions. This transitional regime corresponds to a qualitative change in the shape of the PSNR curve (Figure 1). For $N = 10^5$, the two networks generate nearly identical samples, which no longer resemble images in their corresponding training sets. **Bottom.** The distribution of cosine similarity (normalized inner product) between pairs of images generated by the two networks (blue) shifts from left to right with increasing $N$, showing vanishing model variance. Conversely, the distribution of cosine similarity between generated samples and the most similar image in their corresponding training set (orange) shifts from right to left. For comparison, Appendix B shows the distribution of cosine similarities of closest pairs between the two training subsets, and additional results on the LSUN bedroom dataset (Yu et al., 2015) and for the BF-CNN architecture (Mohan* et al., 2020).

follows that the denoiser mapping can be rewritten in terms of the Jacobian eigendecomposition as

$$f(y) = \nabla f(y)\, y = \sum_k \lambda_k(y)\, \langle y, e_k(y) \rangle\, e_k(y). \tag{6}$$

The denoiser can thus be interpreted as performing shrinkage with factors $\lambda_k(y)$ along axes of a basis specified by $e_k(y)$. Note that both the eigenvalues and eigenvectors depend on the noisy image $y$ (i.e., both the basis and shrinkage factors are *adaptive* (Milanfar, 2012)).

Even if the denoiser is not bias-free, small eigenvalues $\lambda_k(y)$ reveal local invariances of the denoising function: small perturbations in the noisy input along the corresponding eigenvectors $e_k(y)$ do not affect the denoised output. Intuitively, such invariances are a desirable property for a denoiser, and they are naturally enforced by minimizing mean squared error (MSE) as expressed with Stein's unbiased risk estimate (SURE, proved in Appendix D.3 for completeness):

$$\mathrm{MSE}(f, \sigma^2) = \mathbb{E}_y \Big[ 2\sigma^2 \operatorname{tr} \nabla f(y) + \|y - f(y)\|^2 - \sigma^2 d \Big]. \tag{7}$$

To minimize MSE, the denoiser must trade off the approximate "rank" of the Jacobian (the trace is the sum of the eigenvalues) against an estimate of the denoising error: $\|y - f(y)\|^2 - \sigma^2 d$. The denoiser thus locally behaves as a (soft) projection on a subspace whose dimensionality corresponds to the

rank of the Jacobian. As we now explain, this subspace approximates the support of the posterior distribution $p(x|y)$, and thus gives a local approximation of the support of $p(x)$.

It is shown in Appendix D.1 that the optimal minimum MSE denoiser and its Jacobian are given by

$$f^\star(y) = y + \sigma^2 \nabla \log p_\sigma(y) = \mathbb{E}_x[x|y], \tag{8}$$

$$\nabla f^\star(y) = \mathrm{Id} + \sigma^2 \nabla^2 \log p_\sigma(y) = \sigma^{-2} \mathrm{Cov}[x \,|\, y]. \tag{9}$$

That is, the Jacobian of the optimal denoiser is proportional to the posterior covariance matrix, which is symmetric and non-negative. This gives us another interpretation of the adaptive eigenvector basis as providing an optimal approximation of the unknown clean image $x$ given the noisy observation $y$. Further, the optimal denoising error is then given by (see Appendix D.1 for the first equality)

$$\mathrm{MSE}(f^\star, \sigma^2) = \mathbb{E}_y[\mathrm{tr}\,\mathrm{Cov}[x \,|\, y]] = \sigma^2 \,\mathbb{E}_y\big[\mathrm{tr}\,\nabla f^\star(y)\big] = \sigma^2 \,\mathbb{E}_y\left[\sum_k \lambda_k^\star(y)\right]. \tag{10}$$

A small denoising error thus implies an approximately low-rank Jacobian (with many small eigenvalues) and thus an efficient approximation of $x$ given $y$.

In most cases, the optimal adaptive basis $(e_k^\star(y))_{1 \le k \le d}$ is not known. Rather than aiming for exact optimality, classical analyses (Donoho, 1995) thus focus on the asymptotic decay of the denoising error as the noise level $\sigma^2$ falls, up to multiplicative constants. This corresponds to finding a basis $(e_k(y))_{1 \le k \le d}$ which captures the asymptotic slope of the PSNR plots in Figure 1 but not necessarily the intercept. This weaker notion of optimality is obtained by showing matching upper and lower-bounds on the asymptotic behavior of the denoising error. To provide intuition, we first consider a fixed orthonormal basis $e_k(y) = e_k$, and then consider the more general case of best bases selected from a fixed dictionary.

**Denoising in a fixed basis.** Consider a denoising algorithm that is restricted to operate in a fixed basis $e_k$ but can adapt its shrinkage factors $\lambda_k(y)$. An unreachable lower-bound on the denoising error—and thus an upper-bound on the PSNR slope—is obtained by evaluating the performance of an "oracle" denoiser where the shrinkage factors $\lambda_k$ depend on the unknown clean image $x$ rather than the noisy observation $y$ (Mallat, 2008). Appendix D.4 shows that the denoising error of this oracle is

$$\mathbb{E}_x\left[\sum_k \Big((1 - \lambda_k(x))^2 \langle x, e_k\rangle^2 + \lambda_k(x)^2 \sigma^2\Big)\right], \tag{11}$$

which is minimized when $\lambda_k(x) = \frac{\langle x, e_k\rangle^2}{\langle x, e_k\rangle^2 + \sigma^2}$. The coefficient $\lambda_k(x)$ thus acts as a soft threshold: $\lambda_k(x) \approx 1$ when the signal dominates the noise and $\lambda_k(x) \approx 0$ when the signal is weaker than the noise. Appendix D.4 then shows that the oracle denoising error is the expected value of

$$\sigma^2 \sum_k \lambda_k(x) = \sum_k \frac{\sigma^2 \langle x, e_k\rangle^2}{\langle x, e_k\rangle^2 + \sigma^2} \sim \sum_k \min(\langle x, e_k\rangle^2, \sigma^2) = M\sigma^2 + \|x - x_M\|^2, \tag{12}$$

where $x_M = \sum_{\langle x, e_k\rangle^2 > \sigma^2} \langle x, e_k\rangle e_k$ is the $M$-term approximation of $x$ with the $M$ basis coefficients $\langle x, e_k\rangle$ above the noise level, and $\sim$ means that the two terms are of the same order up to multiplicative constants (here smaller than 2). The denoising error is small if $x$ has a sparse representation in the basis, so that both $M$ and the approximation error $\|x - x_M\|^2$ are small. For example, if the coefficients decay as $\langle x, e_k\rangle^2 \sim k^{-(\alpha+1)}$ (up to reordering), Appendix D.4 shows that

$$M\sigma^2 + \|x - x_M\|^2 \sim \sigma^{2\alpha/(\alpha+1)}, \tag{13}$$

which is a lower bound on the MSE of any denoising algorithm in the basis $e_k$. Reciprocally, this oracle denoising error is nearly reached with a soft-thresholding estimator that computes the shrinkage factors $\lambda_k(y)$ by comparing $\langle y, e_k\rangle^2$ (rather than $\langle x, e_k\rangle^2$) with a threshold proportional to $\sigma^2$ (Donoho & Johnstone, 1994), and achieves the decay (13) up to a logarithmic factor. The decay (13) of the MSE with decreasing $\sigma$ corresponds to an asymptotic slope of $\alpha/(\alpha+1)$ in the PSNR curve when the input PSNR increases. Thus, a larger sparsity/regularity exponent $\alpha$, which corresponds to a faster decay of the small coefficients of $x$ in the basis $(e_k)_{1 \le k \le d}$, leads to improved denoising performance.

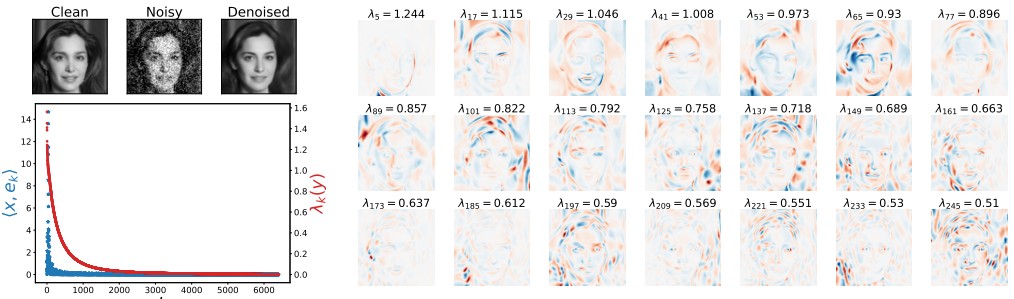

Figure 3: Analysis of a denoiser trained on $10^5$ face images, evaluated on a noisy test image. **Top left.** Clean, noisy ($\sigma = 0.15$) and denoised images. **Bottom left.** Decay of shrinkage values $\lambda_k(y)$ (red), and corresponding coefficients $\langle x, e_k(y) \rangle$ (blue), evaluated for the noisy image $y$. The rapid decay of the coefficients indicates that the image content is highly concentrated within the preserved subspace. **Right.** The adaptive basis vectors $e_k(y)$ contain oscillating patterns, adapted to lie along the contours and within smooth regions of the image, whose frequency increases as $\lambda_k(y)$ decreases.

**Best adaptive bases.** Adapting the basis $(e_k)_{1 \le k \le d}$ to the noisy image $y$ allows obtaining sparser representations of the unknown clean image $x$ with a faster decay, and thus a larger PSNR slope. To calculate the optimal adaptive basis, we need to find an oracle denoiser that has the same asymptotic MSE as a non-oracle denoiser, yielding matching lower and upper bounds on the asymptotic MSE.

Consider an oracle denoiser which performs a thresholding in an oracle basis $(e_k(x))$ that depends on the unknown clean image $x$. The above analysis then still applies, and if the coefficients $\langle x, e_k(x) \rangle^2$ decay as $k^{-(\alpha+1)}$, then the asymptotic PSNR slope is again $\alpha/(\alpha+1)$. The best oracle basis satisfies $e_1(x) = x/\|x\|$, but it yields a loose lower bound as it cannot be estimated from the noisy image $y$ alone. We thus restrict the oracle denoiser to choose the basis $(e_k(x))$ within a fixed dictionary. A larger dictionary increases adaptivity, but it then becomes harder to estimate the basis that best represents $x$ from $y$ alone. If the dictionary of bases is constructed from a number of vectors $e_k$ which is polynomial in the dimension $d$ (the number of bases can however be exponential in $d$) then a thresholding in the basis $(e_k(y))$ that best approximates the noisy image $y$ achieves the same slope as the oracle denoiser (Barron et al., 1999; Dossal et al., 2011). This near-optimality despite the presence of noise comes from the limited choice of possible basis vectors $e_k$ in the dictionary, which limits the variance of the best-basis estimation, e.g. by preventing $e_1(y) = y/\|y\|$. The main difficulty is then to design a small-enough dictionary that gives optimal representations of images from the data distribution in order to achieve the optimal PSNR slope.

We now evaluate the inductive biases of DNN denoisers through this lens. In Section 2, we showed that the DNN denoisers overcome the curse of dimensionality: their variance decays to zero in the generalization regime. In the next section, we explain this observation by demonstrating that they are inductively biased towards adaptive bases $e_k(y)$ from a particular class.

### 3.2 GEOMETRY-ADAPTIVE HARMONIC BASES IN DNNS

Figure 3 shows the shrinkage factors $(\lambda_k(y))$, adaptive basis vectors $(e_k(y))$, and signal coefficients $(\langle x, e_k(y) \rangle)$ of a DNN denoiser trained on $10^5$ face images. The eigenvectors have oscillating patterns both along the contours and in uniformly regular regions and thus adapt to the geometry of the input image. We call this a geometry-adaptive harmonic basis (GAHB). The coefficients are sparse in this basis, and the fast rate of decay of eigenvalues exploits this sparsity. The high quality of generated images and the strong generalization results of Section 2 show that DNN denoisers rely on inductive biases that are well-aligned to photographic image distributions. All of this suggests that DNN denoisers might be inductively biased towards GAHBs. In the following, we provide evidence supporting this conjecture by analyzing networks trained on synthetic datasets where the optimal solution is (approximately) known.

**$C^\alpha$ images and bandlet bases.** If DNNs are inductively biased towards GAHBs, we expect that they generalize and converge to the optimal denoising performance when such bases are optimal. We

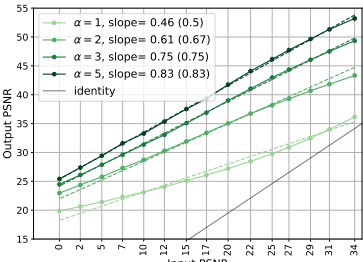 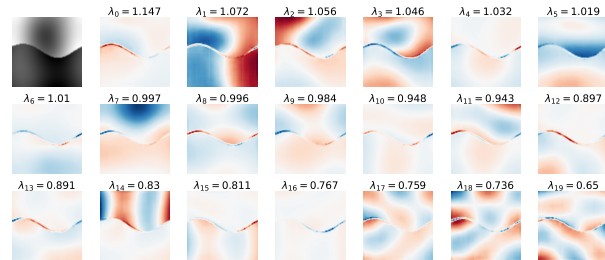

Figure 4: UNet denoisers trained on $10^5$ $\mathbf{C}^\alpha$ images achieve near-optimal performance. **Left.** PSNR curves for various regularity levels $\alpha$. The empirical slopes closely match the theoretical optimal slopes (parenthesized values, dashed lines). **Right.** A $\mathbf{C}^\alpha$ image ($\alpha = 4$) of size $80 \times 80$ and its top eigenvectors, which consist of harmonics on the two regions and harmonics along the boundary. The frequency of the harmonics increases with $k$. More examples are given in Appendix C.1.

consider the so-called geometric $\mathbf{C}^\alpha$ class of images (Korostelev & Tsybakov, 1993; Donoho, 1999; Peyré & Mallat, 2008) which consist of regular contours on regular backgrounds, where the degree of regularity is controlled by $\alpha$. Examples of these images are shown in Figure 4 and Appendix C.1. A mathematical definition and an algorithm for their synthesis are presented in Appendix E.

Optimal sparse representations of $\mathbf{C}^\alpha$ images are obtained with "bandlet" bases (Peyré & Mallat, 2008). Bandlets are harmonic functions oscillating at different frequencies, whose geometry is adapted to the directional regularity of images along contours. Geometric $\mathbf{C}^\alpha$ images can be represented with few bandlets having low-frequency oscillations in regular regions and along contours but sharp variations across contours. The $k$-th coefficient in the best bandlet basis then decays as $k^{-(\alpha+1)}$. It follows that the optimal denoiser has a PSNR which asymptotically increases with a slope $\alpha/(\alpha + 1)$ as a function of input PSNR (Korostelev & Tsybakov, 1993; Dossal et al., 2011).

Figure 4 shows that DNN denoisers trained on $\mathbf{C}^\alpha$ images also achieve this optimal rate and learns GAHBs, similarly to bandlets but with a more flexible geometry. This generalization performance confirms that inductive biases of DNNs favor GAHBs.

**Low-dimensional manifolds.** If DNNs are inductively biased towards GAHBs, then we expect these bases to emerge even in cases where they are suboptimal. To test this prediction, we consider a dataset of disk images with varying positions, sizes, and foreground/background intensities. This defines a five-dimensional *curved* manifold, with a tangent space evaluated at a disk image $x$ that is spanned by deformations of $x$ along these five dimensions. When the noise level $\sigma$ is much smaller than the radius of curvature of the manifold, the posterior distribution $p(x|y)$ is supported on an approximately flat region of the manifold, and the optimal denoiser is approximately a projection onto the tangent space. Thus, the optimal Jacobian should have only five non-negligible eigenvalues, whose corresponding eigenvectors span the tangent space. The remaining eigenvectors should have shrinkage factors of $\lambda = 0$, but are otherwise unconstrained. The optimal MSE is asymptotically equal to $5\sigma^2$, corresponding to a PSNR slope of one.

Figure 5 shows an analysis of a denoiser trained on $10^5$ disk images, of size $80 \times 80$. We observe additional basis vectors with non-negligible eigenvalues that have a GAHB structure, with oscillations on the background region and along the contour of the disk. We also find that the number of non-zero eigenvalues *increases* as the noise level decreases, leading to a suboptimal PSNR slope that is less than $1.0$. These results reveal that the inductive biases of the DNN are not perfectly aligned with low-dimensional manifolds, and that in the presence of the curvature, this suboptimality increases as the noise level decreases. We obtain similar results on two additional examples of a distribution supported on a low-dimensional manifold, given in Appendix C.2.

**Shuffled faces.** We also consider in Appendix C.3 a dataset of shuffled faces, obtained by applying a common permutation to the pixels of each face image. This permutation does not preserve locality between neighboring pixels, and thus the optimal basis does not have harmonic structure. The resulting mismatch between the DNN inductive biases and the data distribution result in substantially worse performance than for the original (unscrambled) faces.

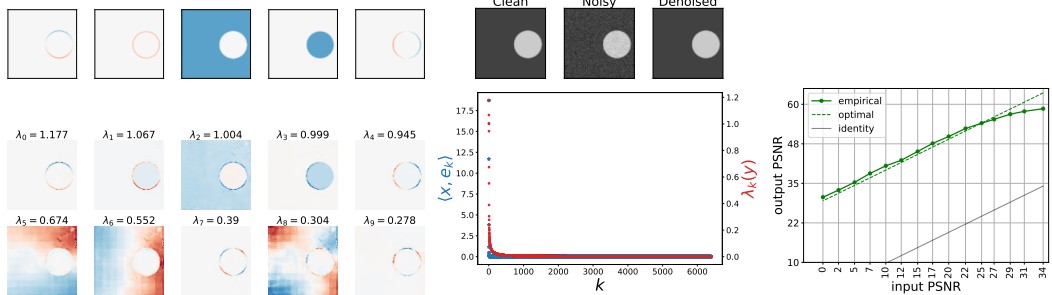

Figure 5: UNet denoiser trained on a dataset of translating and dilating disks, with variable foreground/background intensity. **Top center.** Clean, noisy ($\sigma = 0.04$), and denoised images. **Bottom center.** The decay of shrinkage factors $\lambda_k(y)$ and coefficients $\langle x, e_k(y) \rangle$ indicates that the network achieves and preserves a sparse representation of the true image. **Top right.** denoising performance is sub-optimal, with PSNR slope below the optimal value of $1.0$ for small noise. **Top left.** An optimal basis (in the small-noise limit) spanning the 5-dimensional tangent space of the image manifold. **Bottom left.** Top eigenvectors of the adaptive basis. The first five basis vectors closely match the basis of the tangent space of the manifold evaluated at the clean image. In contrast, the next five are GAHBs that lie along contours and within background regions of the clean image.

## 4    DISCUSSION

Diffusion generative models, which operate through iterative application of a trained DNN denoiser, have recently surpassed all previous methods of learning probability models of images. Their training objective (minimization of squared denoising error) is simple and robust, and they generate samples of impressive quality. In this paper, we elucidate the approximation properties that underlie this success, by analyzing the trained denoiser, which is directly related to the score function, and to the density from which the samples are drawn.

We show empirically that diffusion models memorize samples when trained on small sets, but transition to a strong form of generalization as the training set size increases, converging to a unique density model that is independent of the specific training samples. The amount of data needed to reach this phase transition is very small relative to the size of dataset needed for convergence without any inductive biases, and depends on the image size and complexity relative to the neural network capacity (Yoon et al., 2023). It is of interest to extend both the theory and the empirical studies to account for the interplay of these factors. Appendix B.4 shows preliminary results in this direction.

We also examined the inductive biases that enable this strong generalization. Using a well-established mathematical framework, we showed that DNN denoisers perform shrinkage of noisy coefficients in a geometry-adaptive harmonic basis (GAHB) which is shaped by geometric features of the image. For the $\mathbf{C}^\alpha$ class of images, such geometric bases are known to be optimal, and DNN denoisers achieve near-optimal performance on this class. Previous mathematical literature has shown that bandlet bases, which are a specific type of GAHB, are near-optimal for this class, but the GAHBs learned by the DNN denoiser are more general and more flexible. For images drawn from low-dimensional manifolds, for which the optimal basis spans the tangent subspace of the manifold, we find that DNN denoisers achieve good denoising within a basis aligned with this subspace, but also incorporate GAHB vectors in the remaining unconstrained dimensions. The non-suppressed noise along these additional GAHB components leads to suboptimal denoising performance. This observation, along with similar ones shown in Appendix C.2, provide more supporting evidence for the hypothesis that inductive biases of DNN denoisers promote GAHBs.

We do not provide a formal mathematical definition of the class of GAHBs arising from the inductive biases of DNNs. Convolutions in DNN architectures, whose eigenvectors are sinusoids, presumably engender GAHB harmonic structure, but the geometric adaptivity must arise from interactions with rectification nonlinearities (ReLUs). A more precise elucidation of this GAHB function class, and its role in shaping inductive biases of the DNNs used in a wide variety of other tasks and modalities, is of fundamental interest.

ACKNOWLEDGMENTS

We gratefully acknowledge the support and computing resources of the Flatiron Institute (a research division of the Simons Foundation), and NSF Award 1922658 to the Center for Data Science at NYU.

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

# A EXPERIMENTAL DETAILS

## A.1 TRAINING AND ARCHITECTURE DETAILS

**Architectures.** We performed empirical experiments using two different architectures: UNet, and BF-CNN. All the denoisers are "bias-free": we remove all additive constants from convolution and batch-normalization operations (i.e., the batch normalization does not subtract the mean). This facilitates unversality (denoisers can operate at all noise levels), and interpretability (network transformations are homogeneous of order 1, and the Jacobian provides a local characterization) - see Mohan* et al. (2020).

UNet networks contain 3 decoder blocks, one mid-level block, and 3 decoder blocks (Ronneberger et al., 2015). Each block consists of 2 convolutional layers followed by a ReLU non-linearity and bias-free batch-normalization. Each encoder block is followed by a $2 \times 2$ spacial down-sampling and a 2 fold increase in the number of channels. Each decoder block is followed by a $2 \times 2$ spacial upsampling and a 2 fold reduction of channels. The total number of parameters is $7.6m$.

BF-CNN networks Mohan* et al. (2020) are bias-free versions of DNCNN networks (Zhang et al., 2017), contain 21 convolutional layers with no subsampling, each consisting of 64 channels. Each layer, except for the first and the last, is followed by a ReLU non-linearity and bias-free batch-normalization. All convolutional kernels are of size $3 \times 3$, resulting in $700k$ parameters in total.

**Training.** We follow the training procedure described in Mohan* et al. (2020), minimizing the mean squared error in denoising images corrupted by i.i.d. Gaussian noise with standard deviations drawn from the range $[0, 1]$ (relative to image intensity range $[0, 1]$). Training is carried out on batches of size 512, for 1000 epochs. Note that all denoisers are universal and blind: they are trained to handle a range of noise, and the noise level is not provided as input to the denoiser. These properties are exploited by the sampling algorithm, which can operate without manual specification of the step size schedule (Kadkhodaie & Simoncelli, 2020). This method produces high-quality results in generative sampling, as well as sampling conditioned on linear measurements (Kadkhodaie & Simoncelli, 2021).

**Datasets.** For experiments shown in Figures 1 to 3 and 7, we use the CelebA dataset (Liu et al., 2015) downsampled to $80 \times 80$ resolution. For experiments shown in Figures 9 and 10, we use images drawn from the LSUN bedroom dataset (Yu et al., 2015) downsampled to $80 \times 80$ resolution. This dataset is downsampled to $32 \times 32$ resolution for experiments shown in Figure 13. For experiments shown in Figure 11 we use CelebA HQ dataset (Karras et al., 2018) downsampled to $40 \times 40$ resolution.

## A.2 SAMPLING ALGORITHM

Sampling from both the DNN denoisers is achieved using the algorithm presented in Kadkhodaie & Simoncelli (2020), which is specified below in Algorithm 1. Aside from initial and final noise levels $(\sigma_0, \sigma_\infty)$, this method uses two hyperparameters $h \in [0, 1]$ and $\beta \in (0, 1]$, which control the step size and injected noise respectively. We chose $h = 0.01$, $\beta = 0.1$, $\sigma_0 = 1$, and $\sigma_\infty = 0.05$.

---

**Algorithm 1** Sampling via ascent of the log-likelihood gradient from a denoiser residual

---

**Require:** denoiser $f$, step size $h$, stochasticity from injected noise $\beta$, initial noise level $\sigma_0$, final noise level $\sigma_\infty$, distribution mean $m$

1: $t = 0$
2: Draw $x_0 \sim \mathcal{N}(m, \sigma_0^2 \mathrm{Id})$
3: **while** $\sigma_t \geq \sigma_\infty$ **do**
4: $\quad t \leftarrow t + 1$
5: $\quad s_t \leftarrow f(x_{t-1}) - x_{t-1}$ $\qquad\qquad\qquad$ ▷ Compute the score from the denoiser residual
6: $\quad \sigma_t^2 \leftarrow \|s_t\|^2 / d$ $\qquad\qquad\qquad$ ▷ Compute the current noise level for stopping criterion
7: $\quad \gamma_t^2 = \left((1 - \beta h)^2 - (1 - h)^2\right)\sigma_t^2$
8: $\quad$ Draw $z_t \sim \mathcal{N}(0, I)$
9: $\quad \boldsymbol{x}_t \leftarrow \boldsymbol{x}_{t-1} + h\boldsymbol{d}_t + \gamma_t z_t$ $\qquad\qquad$ ▷ Perform a partial denoiser step and add noise
10: **end while**
11: **return** $x_t$

---

# B    ADDITIONAL NUMERICAL RESULTS ON GENERALIZATION

## B.1    SIMILARITY BETWEEN DATA SUBSETS

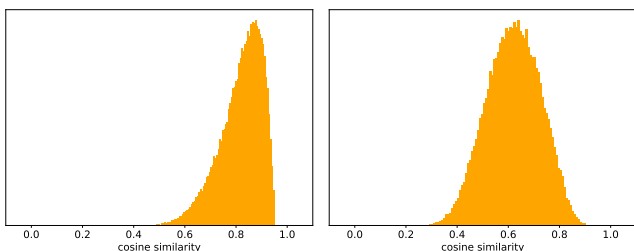

Figure 6: Histogram of cosine similarity between pairs of closest images in the non-overlapping subsets $S_1$ and $S_2$ of CelebA (left) and LSUN bedroom (right). Images with similarity score higher than 0.95 are removed from the datasets before training to eliminate replicated images. This should be compared with the histograms in Figures 2 and 10.

## B.2    GENERALIZATION OF UNET MODEL

In this section, we show that convergence of model variance is robust to the change of data distribution and architecture. The minimum size of the training set, $N$, for which the model transitions from memorization to generalization indeed depends on the architecture, image size and data distribution. Nevertheless, with enough data, two models trained on non-overlapping subsets of data converge to virtually the same function.

### B.2.1    TRAINED ON CELEBA DATASET

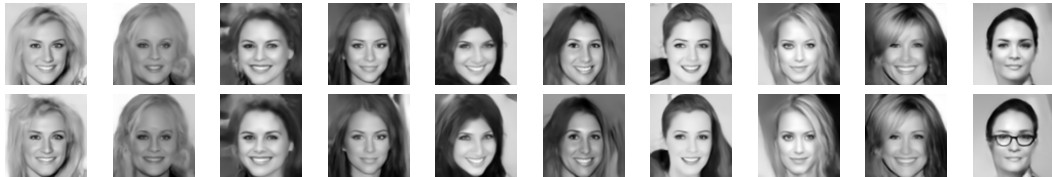

Figure 7: More examples to illustrate convergence of model variance for models shown in Figure 2, at $N = 10^5$. Samples generated by each denoiser are shown in separate rows, where each column shows same initialization across the networks. The networks generate nearly identical samples, showing convergence to the same function.

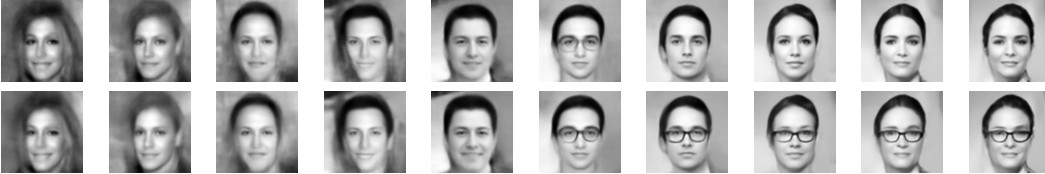

Figure 8: Bifurcation of trajectories. Sampling trajectories for the two samples shown in the last column of Figure 7. The two diffusion models arrive at different samples starting from the same initial point. The bifurcation of gradients appears to emerge somewhere around the middle of the trajectories, which illustrates instabilities predicted by recent dynamical models (Biroli et al., 2024). All the intermediate samples in the trajectories have been denoised in a on-shot denoising manner using the corresponding denoisers. This example shows that the convergence is not perfect, hence the distribution of cosine similarities at $N = 10^5$ is not perfectly a delta function at 1.

### B.2.2 TRAINED ON LSUN BEDROOM DATASET

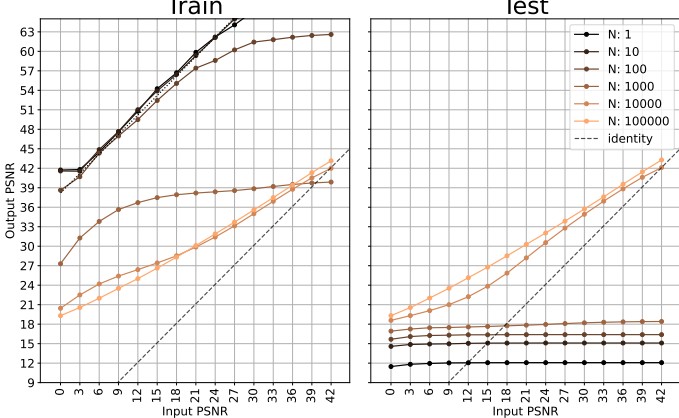

Figure 9: Transition from memorization to generalization, for a UNet denoiser trained on bedroom LSUN images (Yu et al., 2015) downsampled to $80 \times 80$. Similarly to denoisers trained on face images shown in Figure 1, the model transitions from memorizing the training set to generalizing outside of the training set. At $N = 10^5$ the performance is almost identical on training and test sets, and the model is no longer overfitting the training data.

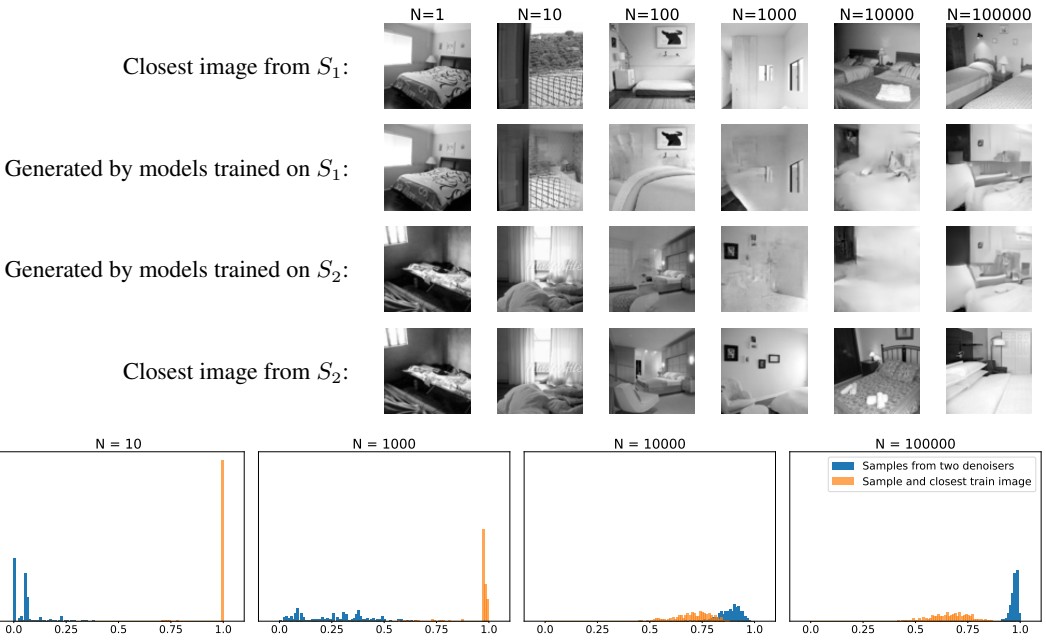

Figure 10: Convergence of model variance. Diffusion models are trained on non-overlapping subsets $S_1$ and $S_2$ of a bedroom LSUN dataset. The subset size $N$ varies from 1 to $10^5$. Notice the samples generated by network trained on $N = 100$ images: they are combinations of patches of training images. This type of memorization has been previously reported in (Somepalli et al., 2023). See caption of Figure 2 for a complete description of the figure.

### B.3 GENERALIZATION OF BF-CNN MODEL

#### B.3.1 TRAINED ON CELEBA DATASET

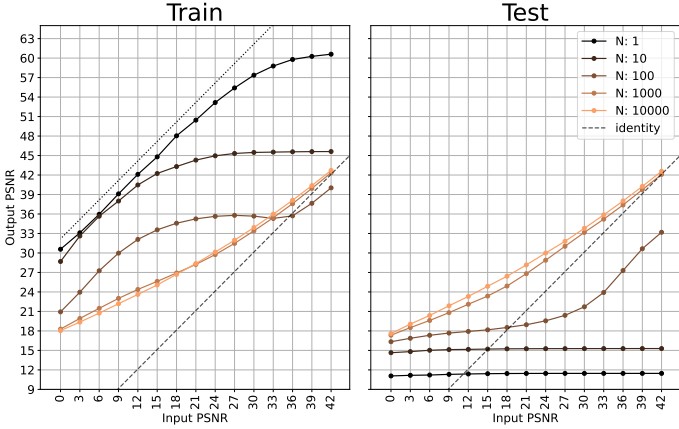

Figure 11: Transition from memorization to generalization, for a BF-CNN denoiser trained on CelebA HQ dataset (Karras et al., 2018) downsampled to $40 \times 40$ resolution. See caption of Figure 1.

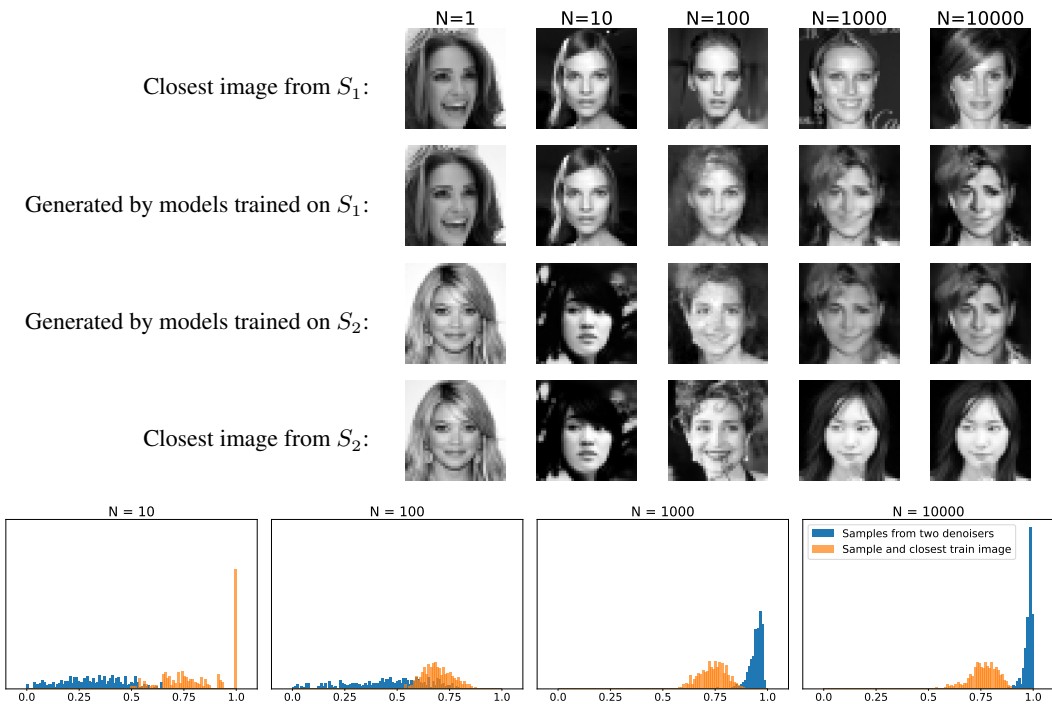

Figure 12: Convergence of model variance. BF-CNN denoisers are trained on non-overlapping subsets $S_1$ and $S_2$ of CelebA HQ dataset. The subset size $N$ varies from 1 to $10^4$. See caption of Figure 2.

### B.3.2 TRAINED ON LSUN BEDROOM DATASET

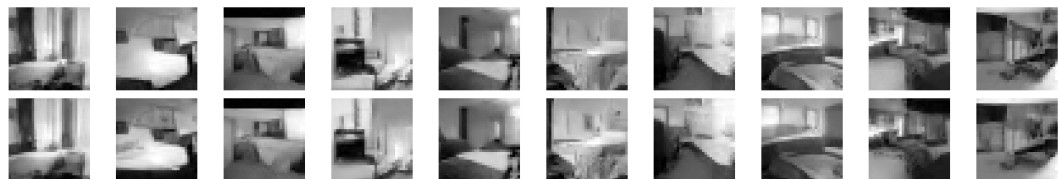

Figure 13: Convergence of model variance on LSUN bedroom dataset (Yu et al., 2015). A dataset of bedroom images is partitioned into two non-overlapping datasets, $S_1$ and $S_2$, each containing $N = 20,000$ images down-sampled to size $32 \times 32$. We train two networks (BF-CNN architecture described in Appendix A) on $S_1$ and $S_2$. Each network is then used in an iterative deterministic reverse diffusion algorithm to generate a sample, with both networks initialized with the same noise image. Samples generated by each denoiser are shown in separate rows, where each column shows same initialization across the networks. The networks generate nearly identical samples, showing convergence to the same function.

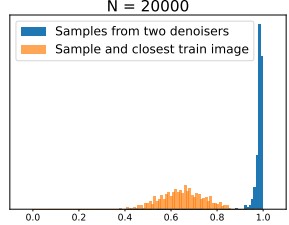

Figure 14: Blue histogram: cosine similarity between samples generated by two denoisers trained on non-overlapping training sets of size $N = 20,000$ from LSUN bedroom dataset downsampled to $32 \times 32$ resolution. Orange histograms: cosine similarity between generated samples and the closest image from the corresponding training set. Images drawn from the two denoisers are very similar to each other, compared to the closest image in their respective training sets.

### B.4 Convergence as a function of training set size $N$ and image resolution

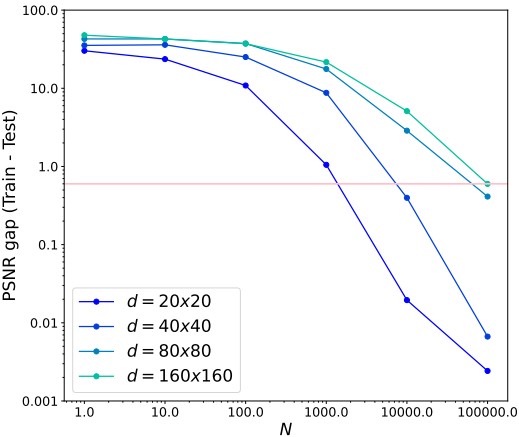

Figure 15: Generalization as a function of training set size, $N$. Generalization is measured as the difference between train and test PSNRs averaged across noise levels $\sigma \in [0, 1]$. Each curve shows average PSNR gap for a specific image resolution $d \times d$. The capacity of the UNet is adjusted to the image resolution. See Table 1 for specific UNet architecture used for each image resolution. As expected, to reach the threshold PSNR gap, denoisers trained on larger images require more training data. However, the number of training images does not increase proportionally with the image size: the increase in needed to hit the threshold $\Delta N$ from $80 \times 80$ to $160 \times 160$ images is much smaller than the $\Delta N$ from $40 \times 40$ to $80 \times 80$. This observation is consistent with previous reports indicating that conditioned on coarser content of the image, learning the finer details requires less data due to the conditional Markov property of images (Kadkhodaie et al., 2023).

| Image resolution | number of encoder decoder blocks | Receptive field size | number of parameters |
|---|---|---|---|
| $20 \times 20$ | 1 | $18 \times 18$ | $360k$ |
| $40 \times 40$ | 2 | $44 \times 44$ | $1.8m$ |
| $80 \times 80$ | 3 | $92 \times 92$ | $7.6m$ |
| $160 \times 160$ | 4 | $188 \times 188$ | $31m$ |

Table 1: UNet architectures used in experiments shown in Figure 15. With the four-fold increase of the image size, the number of parameters increases approximately four times.

# C  ADDITIONAL NUMERICAL RESULTS ON INDUCTIVE BIASES

## C.1  MORE $\mathbf{C}^\alpha$ EXAMPLES

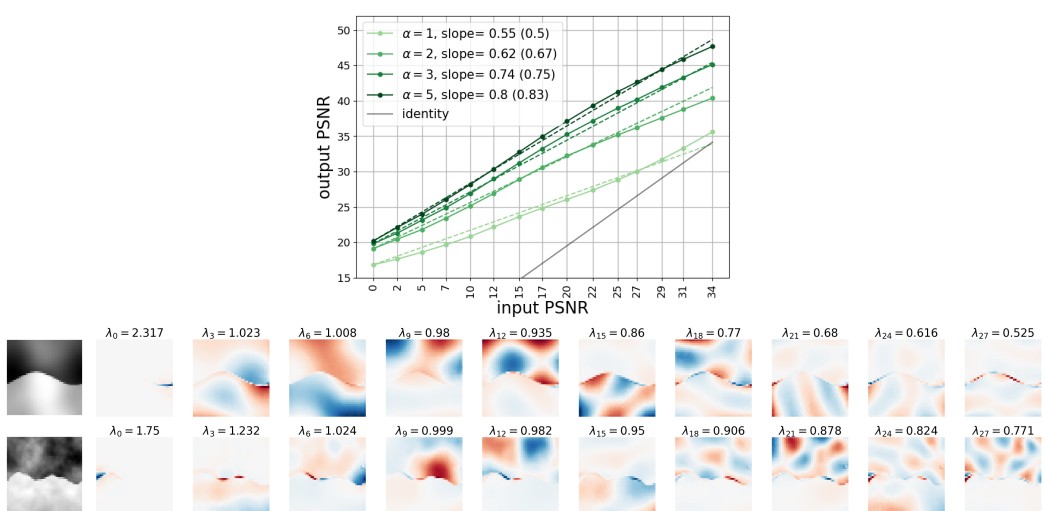

Figure 16: BF-CNN denoisers trained on $\mathbf{C}^\alpha$ images of size $40\times40$ achieve near-optimal performance. **Top.** PSNR curves of trained networks for various regularity levels $\alpha$. The empirical slopes achieved for different values of $\alpha$ closely match the optimal slopes (dashed lines). **Bottom.** Eigenvectors for two $\mathbf{C}^\alpha$ images (top row: $\alpha = 4$, bottom row: $\alpha = 2$), which consist of harmonics on the two regions and harmonics along the boundary. The frequency of the harmonics increases with $k$. For less regular images, the harmonics are more localized along the contours.

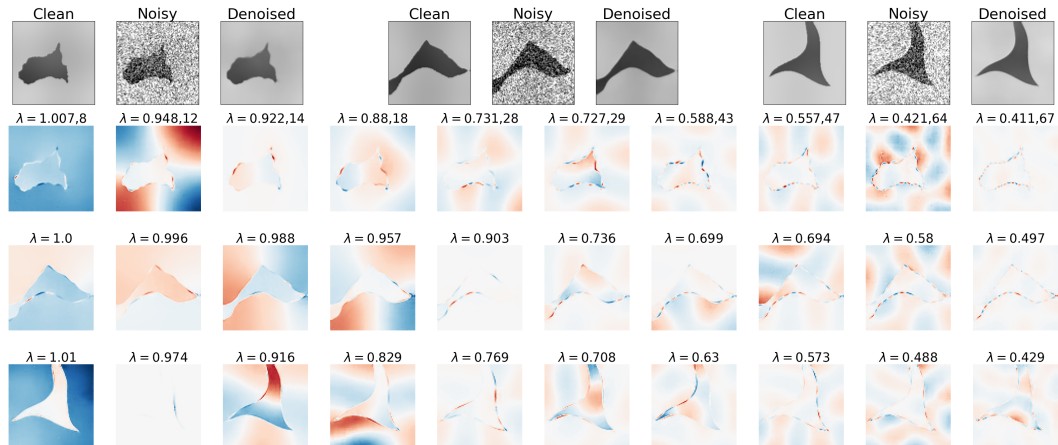

Figure 17: Geometric-adaptive harmonic basis shown for three test images from $\mathbf{C}^\alpha$ class. Here the regularity of the one-dimensional contours $\alpha_1$ is different from the regularity of the two-dimensional background $\alpha_2$. **Top.** Three example images. The regularity of the contour increases from left to right: $\alpha_1 = 1.5, 2, 4$. Background regularity is the same in all three examples, $\alpha_2 = 8$, and $\sigma = 0.2$. **Bottom.** Top 10 basis vectors for each image are shown. With increasing $\alpha_1$, the contours become more regular, and the harmonics along the boundaries become less localized. This allows for a faster decay of coefficients and a lower denoising error.

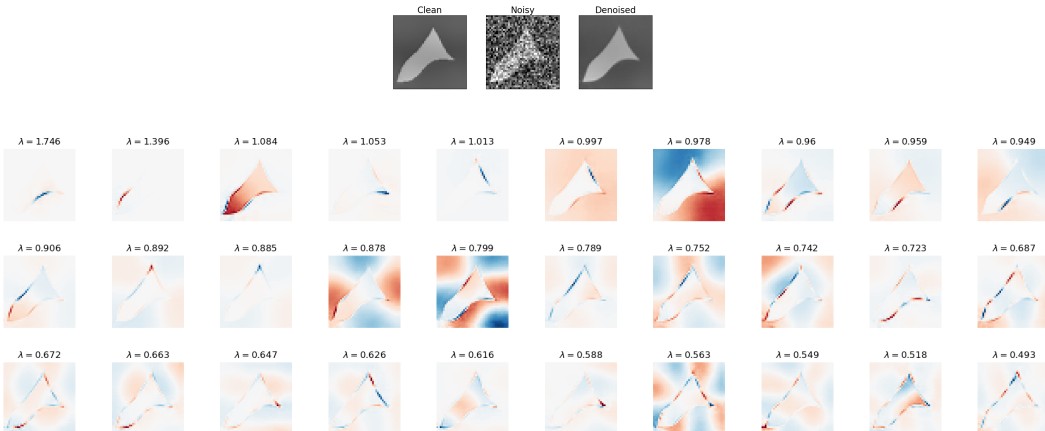

Figure 18: **Top.** An additional example of a $C^\alpha$ test image with $\alpha = 3$. **Bottom.** Top eigenvectors of the geometric harmonic adaptive basis.

## C.2 ADDITIONAL LOW-DIMENSIONAL MANIFOLD EXAMPLES

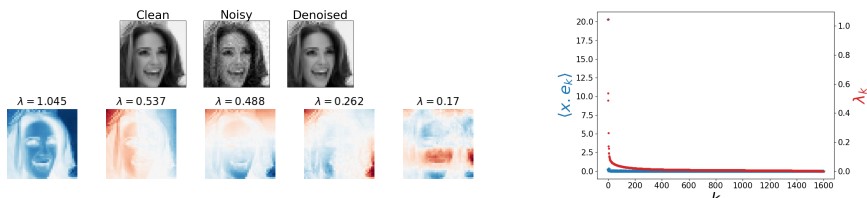

Figure 19: BF-CNN denoiser trained on a single face image, with intensity rescaling. We consider an image class consisting of a single image $x \in \mathbb{R}^d$ and its positive rescalings $s\,x$ for $s > 0$. The resulting images lie on a ray emanating from the origin, and optimal denoising corresponds to projecting the noisy image onto this ray. The optimal denoising basis should therefore include the normalized vector $x/\|x\|$ with associated shrinkage factor $\lambda = 1$, whereas the remaining basis vectors should have shrinkage factors of $\lambda = 0$ but are otherwise unconstrained. This optimal denoiser achieves an MSE of $\sigma^2$, and thus a linear PSNR curve with unit slope and intercept $10 \log_{10}(d)$. **Top left**. Denoising of the training image with $\sigma = 0.04$. **Right**. Decay of the coefficients $\langle x, e_k \rangle$ and the shrinkage factors $\lambda_k$. The DNN denoiser exhibits a slower decay of shrinkage factors than the optimal solution, which results in suboptimal performance. **Bottom left**. Top 5 basis vectors $e_k(y)$. The first basis vector is nearly identical to the (normalized) train image. The next vectors, which have non-zero shrinkage factors, exhibit 2D harmonics. These GAHB components underlie the non-optimal behavior of the denoiser. Specifically, the $N = 1$ curve in the left panel of Figure 11 shows that performance as a function of noise level falls below the optimal solution (dotted line). The DNN performance has a unit slope over most of the noise range but has a less-than-optimal intercept (the flattening of the curve at small noise levels is a result of de-emphasis of small noise levels during training).

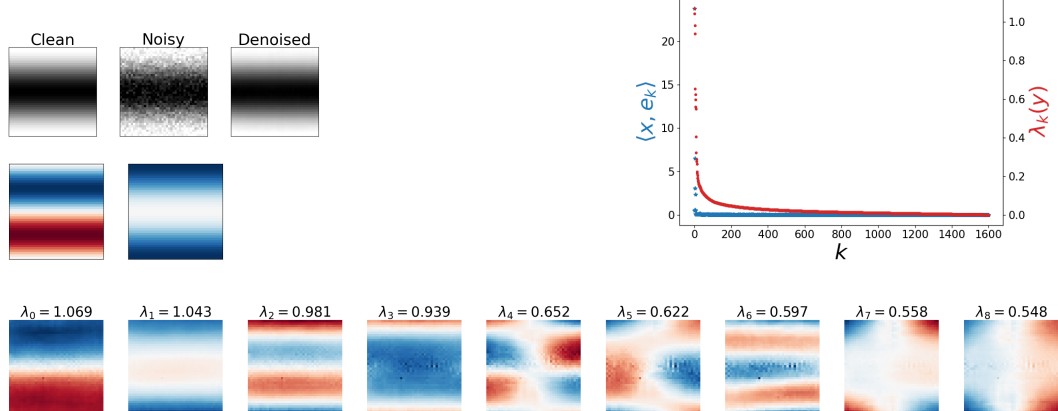

Figure 20: A BF-CNN denoiser is trained on a set of 2D sine wave images with unit frequency and varying phases and intensities. The train images thus lie on a 2D cone manifold with low curvature. For small $\sigma$, the manifold can be assumed to be locally flat, so that the optimal denoising is achieved by projecting the noisy image on the two-dimensional subspace tangent to the manifold. This subspace is spanned by two sine waves with unit frequency and a $\pi/2$ phase shift. **Top left.** Clean, noisy ($\sigma = 0.08$), and denoised test image. **Middle left.** The unit vectors spanning the tangent subspace. The optimal denoising results from projection onto this subspace. **Bottom row.** Empirical basis obtained from the network Jacobian. The empirical solution has a slower decay than optimal (i.e., $\langle x, e_k(y) \rangle > 0$ for $k \geq 2$, as seen in the **right** panel), with harmonic patterns. This sub-optimality reveals the nature of the inductive bias.

## C.3    SHUFFLED FACES

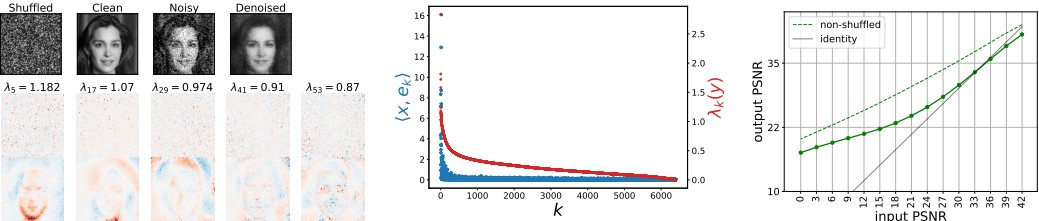

Figure 21: DNN denoiser trained on a dataset of shuffled faces obtained by permuting the pixels of $10^5$ face images in the CelebA dataset. The permutation was chosen randomly, and does not preserve locality, as neighboring pixels are mapped to independent positions. By construction, the optimal denoiser on shuffled faces has the same performance as the optimal denoiser on ordinary faces (unshuffling the image pixels, optimally denoising the face image, and then shuffling the pixels back). For visualization purposes, we "unshuffle" the pixels by applying the inverse of the permutation to the images before display. **Top left.** Clean (shuffled then unshuffled), noisy (unshuffled, $\sigma = 0.3$), and denoised (unshuffled) images. **Middle.** The shrinkage factors $\lambda_k(y)$ decay more slowly than when the denoiser is trained on non-shuffled faces (Figure 3), which is indicative of suboptimality.. **Right.** The denoiser performs significantly worse than the denoiser trained on unshuffled faces: the MSE is much higher with a much lower PSNR slope. **Bottom left.** Basis vectors (top row: shuffled, bottom row: unshuffled). After unshuffling, we observe GAHBs adapted to the geometry of the face, although these are noisier and less precisely aligned with the image features than the non-shuffled examples in Figure 3.

# D MATHEMATICAL DERIVATIONS

## D.1 MIYASAWA RELATIONSHIPS

The relationship of MMSE estimation of a signal corrupted by additive Gaussian noise to the score was published in Miyasawa (1961), and generalized in Raphan & Simoncelli (2011); Efron (2011). For completeness, and notational consistency, we provide a derivation here. We begin by expressing the score $\nabla \log p(y)$ (dropping the $\sigma$ dependence to simplify notation) and its Jacobian $\nabla^2 \log p(y)$ in terms of the measurement density $p(y|x)$ (which is Gaussian) and the posterior density $p(x|y)$. Using Bayes' rule and marginalization, the probability density of the noisy images is expressed as

$$p(y) = \int p(x)\, p(y|x)\, \mathrm{d}x.$$

Taking the logarithm and differentiating with respect to $y$, and using the fact that for any function $h$, $\nabla h(y) = h(y)\, \nabla \log h(y)$, we find

$$\nabla \log p(y) = \int p(x)\, p(y|x)\, \nabla_y \log p(y|x)\, \mathrm{d}x \bigg/ p(y)$$

$$= \int p(x|y)\, \nabla_y \log p(y|x)\, \mathrm{d}x$$

$$= \mathbb{E}\big[\nabla_y \log p(y|x)\,\big|\, y\big], \tag{14}$$

which can be thought of as an equivalent of the chain rule on the scores as opposed to the densities.

Differentiating again with respect to $y$, we have

$$\nabla^2 \log p(y) = \int p(x|y)\Big(\nabla_y \log p(x|y)\nabla_y \log p(y|x)^{\mathrm{T}} + \nabla^2 \log p(y|x)\Big)\mathrm{d}x. \tag{15}$$

The term $\nabla_y \log p(x|y)$ can be calculated by differentiating the logarithm of Bayes rule:

$$\log p(x|y) = \log p(y|x) - \log p(y) + \log p(x),$$
$$\nabla_y \log p(x|y) = \nabla_y \log p(y|x) - \nabla \log p(y), \tag{16}$$

so that when injected into eq. (15) we obtain

$$\nabla^2 \log p(y) = \int p(x|y)\Big(\big(\nabla_y \log p(y|x) - \nabla \log p(y)\big)\nabla_y \log p(y|x)^{\mathrm{T}} + \nabla^2 \log p(y)\Big)\mathrm{d}x$$

$$= \mathbb{E}\Big[\big(\nabla_y \log p(y|x) - \nabla \log p(y)\big)\nabla_y \log p(y|x)^{\mathrm{T}}\,\Big|\, y\Big] + \mathbb{E}\Big[\nabla^2 \log p(y|x)\,\Big|\, y\Big]$$

$$= \mathrm{Cov}\big[\nabla_y \log p(y|x)\,\big|\, y\big] + \mathbb{E}\Big[\nabla^2 \log p(y|x)\,\Big|\, y\Big], \tag{17}$$

where the last line used $\nabla \log p(y) = \mathbb{E}\big[\nabla_y \log p(y|x)\,\big|\, y\big]$.

We then use the fact that $y$ is obtained from $x$ by adding Gaussian white noise of variance $\sigma^2 \mathrm{Id}$:

$$\log p(y|x) = -\frac{1}{2\sigma^2}\|y - x\|^2 + \mathrm{cst}, \tag{18}$$

$$\nabla_y \log p(y|x) = -\frac{1}{\sigma^2}(y - x), \tag{19}$$

$$\nabla_y^2 \log p(y|x) = -\frac{1}{\sigma^2}\mathrm{Id}, \tag{20}$$

so that eqs. (14) and (17) become

$$\nabla \log p(y) = \frac{1}{\sigma^2}(\mathbb{E}[x\,|\,y] - y),$$

$$\nabla^2 \log p(y) = \frac{1}{\sigma^4}\mathrm{Cov}[x\,|\,y] - \frac{1}{\sigma^2}\mathrm{Id}.$$

Finally, the above identities can be rearranged to yield the first- and second-order Miyasawa relationships:

$$\mathbb{E}[x \mid y] = y + \sigma^2 \nabla \log p(y), \tag{21}$$

$$\mathrm{Cov}[x \mid y] = \sigma^2 \Big( \mathrm{Id} + \sigma^2 \nabla^2 \log p(y) \Big). \tag{22}$$

Note that the optimal denoising error satisfies

$$\mathbb{E}\Big[\|x - \mathbb{E}[x \mid y]\|^2\Big] = \mathbb{E}\Big[\mathbb{E}\Big[\mathrm{tr}(x - \mathbb{E}[x \mid y])(x - \mathbb{E}[x \mid y])^{\mathrm{T}} \;\Big|\; y\Big]\Big] = \mathbb{E}[\mathrm{tr}\,\mathrm{Cov}[x \mid y]].$$

## D.2 CONTROL ON KULLBACK-LEIBLER DIVERGENCE

Equation (2) results from Theorem 1 of Song et al. (2021), considering the so-called "variance-exploding" SDE $\mathrm{d}x_t = \mathrm{d}w_t$ where $(w_t)_{t \geq 0}$ is a Brownian motion ($t = \sigma^2$ then corresponds to the noise variance), and letting the stopping time $T$ go to infinity.

To reformulate the score-matching error as a denoising objective, we insert the Miyasawa equation (3) as well as the expression of the score model $s_\theta(y) = (f_\theta(y) - y)/\sigma^2$ into the score-matching error:

$$\mathbb{E}\Big[\|\nabla \log p_\sigma(y) - s_\theta(y)\|^2\Big] = \frac{1}{\sigma^4}\mathbb{E}\Big[\|\mathbb{E}[x \mid y] - f_\theta(y)\|^2\Big]. \tag{23}$$

We recall the decomposition of the denoising error when conditioning on $y$:

$$\mathbb{E}\Big[\|x - f_\theta(y)\|^2\Big] = \mathbb{E}\Big[\|x - \mathbb{E}[x \mid y]\|^2\Big] + \mathbb{E}\Big[\|\mathbb{E}[x \mid y] - f_\theta(y)\|^2\Big], \tag{24}$$

so that inserting eq. (24) into eq. (23) yields

$$\begin{aligned}
\mathbb{E}\Big[\|\nabla \log p_\sigma(y) - s_\theta(y)\|^2\Big] &= \frac{1}{\sigma^4}\Big(\mathbb{E}\Big[\|x - f_\theta(y)\|^2\Big] - \mathbb{E}\Big[\|x - \mathbb{E}[x \mid y]\|^2\Big]\Big) \\
&= \frac{1}{\sigma^4}\Big(\mathrm{MSE}(f_\theta, \sigma^2) - \mathrm{MSE}(f^\star, \sigma^2)\Big).
\end{aligned}$$

Combined with eq. (2), this proves eq. (5).

## D.3 SURE OBJECTIVE

We decompose the MSE as follows:

$$\begin{aligned}
\mathbb{E}\Big[\|x - f(y)\|^2\Big] &= \mathbb{E}\Big[\|(y - f(y)) - (y - x)\|^2\Big] \\
&= \mathbb{E}\Big[\|y - f(y)\|^2\Big] - 2\mathbb{E}[\langle y - x, y - f(y)\rangle] + \mathbb{E}\Big[\|y - x\|^2\Big]. \tag{25}
\end{aligned}$$

The last term is the total variance of the noise and is thus equal to $\sigma^2 d$. The middle term can be rewritten with an integration by parts, using the fact that $y - x = -\sigma^2 \nabla_y \log p(y|x)$:

$$\begin{aligned}
\mathbb{E}[\langle y - x, y - f(y)\rangle] &= -\sigma^2 \iint \langle \nabla_y \log p(y|x), y - f(y)\rangle\, p(x)\, p(y|x)\, \mathrm{d}x\, \mathrm{d}y, \\
&= -\sigma^2 \iint \langle \nabla_y p(y|x), y - f(y)\rangle\, p(x)\, \mathrm{d}x\, \mathrm{d}y, \\
&= \sigma^2 \iint \mathrm{tr}(\mathrm{Id} - \nabla f(y))\, p(x)\, p(y|x)\, \mathrm{d}x\, \mathrm{d}y, \\
&= \sigma^2 \mathbb{E}[d - \mathrm{tr}\,\nabla f(y)]. \tag{26}
\end{aligned}$$

Inserting eq. (26) into eq. (25), we then obtain

$$\mathbb{E}\Big[\|x - f(y)\|^2\Big] = \mathbb{E}\Big[\|y - f(y)\|^2\Big] + 2\sigma^2 \mathbb{E}[\mathrm{tr}\,\nabla f(y)] - \sigma^2 d, \tag{27}$$

proving the Stein's Unbiased Risk Estimator of the MSE.

### D.4 OPTIMAL THRESHOLDING IN A BASIS

For completeness, we derive here the error of the fixed-basis oracle denoiser (Donoho & Johnstone, 1994; Donoho, 1995; Mallat, 2008).

We consider an oracle denoiser which computes

$$\sum_k \lambda_k(x) \langle y, e_k \rangle e_k.$$

In practice, the denoiser does not have access to the clean image $x$, and the shrinkage factors $\lambda_k$ thus have to be estimated from the noisy image $y$ alone. Note however that optimizing this oracle estimator is non-trivial as the shrinkage factors have to be independent from the noise.

We can then compute its denoising error on a clean image $x$ by averaging over the noise

$$\mathbb{E}\left[\left\|x - \sum_k \lambda_k(x) \langle y, e_k \rangle e_k\right\|^2 \Big| x\right] = \mathbb{E}\left[\sum_k (\langle x, e_k \rangle - \lambda_k(x) \langle y, e_k \rangle)^2 \Big| x\right]$$

$$= \mathbb{E}\left[\sum_k ((1 - \lambda_k(x))\langle x, e_k \rangle - \lambda_k(x)\langle z, e_k \rangle)^2 \Big| x\right]$$

$$= \sum_k \left((1 - \lambda_k(x))^2 \langle x, e_k \rangle^2 + \lambda_k(x)^2 \sigma^2\right), \qquad (28)$$

where the last step used the fact that $\langle z, e_k \rangle \sim \mathcal{N}(0, \sigma^2)$ independently from $x$. For each $x$ and $k$, the optimal oracle shrinkage factor $\lambda_k(x)$ thus minimizes the quadratic function

$$(1 - \lambda_k(x))^2 \langle x, e_k \rangle^2 + \lambda_k(x)^2 \sigma^2,$$

which is achieved when

$$\lambda_k(x) = \frac{\langle x, e_k \rangle^2}{\langle x, e_k \rangle^2 + \sigma^2}. \qquad (29)$$

Injecting eq. (29) into eq. (28) gives the denoising error on $x$ as

$$\mathbb{E}\left[\left\|x - \sum_k \lambda_k(x) \langle y, e_k \rangle e_k\right\|^2 \Big| x\right] = \sum_k \frac{\sigma^2 \langle x, e_k \rangle^2}{\langle x, e_k \rangle^2 + \sigma^2}. \qquad (30)$$

Incidentally, this error is also equal to $\sigma^2 \sum_k \lambda_k(x)$, similarly to the optimal denoiser as shown in eq. (10).

The fraction $\frac{\sigma^2 \langle x, e_k \rangle^2}{\langle x, e_k \rangle^2 + \sigma^2}$ is of the same order as $\min(\langle x, e_k \rangle^2, \sigma^2)$ up to a factor of 2, as we have the inequalities for any $a, b > 0$

$$\frac{1}{2} \min(a, b) \le \frac{ab}{a + b} \le \min(a, b),$$

which follow from $ab = \min(a, b) \max(a, b)$ and $\max(a, b) \le a + b \le 2 \max(a, b)$. We thus have

$$\mathbb{E}\left[\left\|x - \sum_k \lambda_k(x) \langle y, e_k \rangle e_k\right\|^2 \Big| x\right] \sim \sum_k \min\left(\langle x, e_k \rangle^2, \sigma^2\right)$$

$$= \sum_{\langle x, e_k \rangle^2 > \sigma^2} \sigma^2 + \sum_{\langle x, e_k \rangle^2 < \sigma^2} \langle x, e_k \rangle^2. \qquad (31)$$

Let $M$ be the number of terms in the left sum (that is, the number of ranks $k$ such that $\langle x, e_k \rangle^2 > \sigma^2$), and $x_M = \sum_{\langle x, e_k \rangle^2 > \sigma^2} \langle x, e_k \rangle e_k$ be the $M$-term approximation of $x$. We then have

$$\|x - x_M\|^2 = \left\|\sum_{\langle x, e_k \rangle^2 < \sigma^2} \langle x, e_k \rangle e_k\right\| = \sum_{\langle x, e_k \rangle^2 < \sigma^2} \langle x, e_k \rangle^2, \qquad (32)$$

so that plugging eq. (32) into eq. (31) gives

$$\mathbb{E}\left[\left\|x - \sum_k \lambda_k(x)\langle y, e_k\rangle e_k\right\|^2 \,\middle|\, x\right] \sim M\sigma^2 + \|x - x_M\|^2. \tag{33}$$

This realizes a decomposition of the oracle denoising error into a denoising bias $\|x - x_M\|^2$, which corresponds to the signal variance that has been lost, and a denoising variance $M\sigma^2$, which corresponds to the noise variance that has been preserved (note that denoising bias and variance are different than the model variance and model bias studied in the paper). The sum of the two terms captures the efficiency of the approximation of $x$ in the basis $(e_k)$.

Let us reorder the coefficients so that $\langle x, e_1\rangle^2 \geq \cdots \geq \langle x, e_d\rangle^2$ (note that the ordering depends on $x$), and assume that $\langle x, e_k\rangle^2 \sim k^{-(\alpha+1)}$ for some $\alpha > 0$. More precisely, we assume that there exists two constants $c, c'$ independent of $x$ and $k$ such that $c\,k^{-(\alpha+1)} \leq \langle x, e_k\rangle^2 \leq c'\,k^{-(\alpha+1)}$. By definition of $M$,

$$\langle x, e_M\rangle^2 > \sigma^2 \geq \langle x, e_{M+1}\rangle^2,$$

so that

$$c'\,M^{-(\alpha+1)} > \sigma^2 \geq c\,(M+1)^{-(\alpha+1)}.$$

We then have $M^{-(\alpha+1)} \sim \sigma^2$, i.e., $M \sim \sigma^{-2/(\alpha+1)}$, and thus $M\sigma^2 \sim \sigma^{2\alpha/(\alpha+1)}$. We also have

$$\sum_{k>M} \langle x, e_k\rangle^2 \leq c' \sum_{k>M} k^{-(\alpha+1)} \leq c' \int_M^{+\infty} t^{-(\alpha+1)}\mathrm{d}t = \frac{c'}{\alpha}M^{-\alpha},$$

$$\sum_{k>M} \langle x, e_k\rangle^2 \geq c \sum_{k>M} k^{-(\alpha+1)} \geq c \int_{M+1}^{+\infty} t^{-(\alpha+1)}\mathrm{d}t = \frac{c}{\alpha}(M+1)^{-\alpha},$$

so that $\|x - x_M\|^2 \sim M^{-\alpha} \sim \sigma^{2\alpha/(\alpha+1)}$. Finally, we have shown that the two terms in eq. (33) are of the same order, and it follows that

$$\mathbb{E}\left[\left\|x - \sum_k \lambda_k(x)\langle y, e_k\rangle e_k\right\|^2 \,\middle|\, x\right] \sim M\sigma^2 + \|x - x_M\|^2 \sim \sigma^{2\alpha/(\alpha+1)}.$$

Because the constants have been assumed to be independent of $x$, one can average over $x$ to obtain that the oracle MSE is $\sim \sigma^{2\alpha/(\alpha+1)}$.

# E  GEOMETRIC $\mathbf{C}^\alpha$ IMAGES

A continuous image $x\colon [0,1]^2 \to \mathbb{R}$ is part of the geometric $\mathbf{C}^\alpha$ class (Korostelev & Tsybakov, 1993; Donoho, 1999; Peyré & Mallat, 2008) if it is uniformly $\alpha$-Lipschitz over $[0,1]^2 \setminus \{\gamma_i\}$, where the $\gamma_i$ are uniformly $\alpha$-Lipschitz curves in $[0,1]^2$ which do not intersect tangentially. A function $f$ is uniformly $\alpha$-Lipschitz over a domain $\Omega$ if there exists a constant $C$ such that for all $x \in \Omega$, there exists a polynomial $q_x$ of degree $\lfloor\alpha\rfloor$ such that for all $y \in \Omega$,

$$|f(y) - q_x(y)| \leq C\,|x - y|^\alpha. \tag{34}$$

We explain how to generate numerically such images in Algorithm 2.

---

**Algorithm 2** Synthesis of a $\mathbf{C}^\alpha$ image via integration

---

**Require:** regularity $\alpha$, Fast Fourier Transform FFT
1: **Make a contour**
2: Define a 1D filter $f_1(\omega) = |\omega|^{-\alpha}$
3: Draw a random 1D $\mathbf{C}^0$ function with i.i.d. uniform entries $c(t) \sim \mathcal{U}([-0.5, 0.5])$
4: Integrate in the Fourier domain to define $C = \mathrm{FFT}^{-1}(f_1 \times \mathrm{FFT}(c))$
5: **Make the background**
6: Define a 2D filter $f_2(\omega) = (\omega_1^2 + \omega_2^2)^{-\alpha/2}$
7: Draw two random 2D $\mathbf{C}^0$ functions with i.i.d. uniform entries $b_1(x, y), b_2(x, y) \sim \mathcal{U}([-0.5, 0.5])$
8: Integrate in the Fourier domain to define $B_i = \mathrm{FFT}^{-1}(f_2 \times \mathrm{FFT}(b_i))$ $(i = 1, 2)$
9: **Make a mask and combine**
10: Define a binary mask $M = \mathbb{1}_{y > C}$
11: Let $x = M \times B_1 + (1 - M) \times B_2$
12: **return** $x$

---

