# OpenReview forum: "Generalization in diffusion models arises from geometry-adaptive harmonic representations"
_ICLR.cc/2024/Conference — ICLR 2024 oral_

### Official Review · Reviewer_89Fo · 2023-10-25

**Soundness:** 3 good
**Presentation:** 3 good
**Contribution:** 3 good
**Rating:** 8
**Confidence:** 4

**Summary:**

The paper studies the inductive bias of the models which involve a denoising step, i.e., score-based diffusion algorithms. To do so, the authors take a spectral approach and analyze the eigenspace and eigenvalues of the corresponding DNN denoiser Jacobian. The authors conjecture that the DNN denoiser are implicitly biased toward so called geometry-adaptive harmonic bases (GAHBs). The authors first show that such biases emerge on a set of synthetic $C^{\alpha}$ images. Further strengthening a point,  the authors empirically show that the bias still emergences similarly even in suboptimal scenario (disk images and shuffled CelebA).

**Strengths:**

- the paper is easy to follow and elaborate
- a good set experiments which validate the main points of paper fairly, e.g., optimal PSNR for $C^\alpha$ images and slow eigenvalues decay on CelebA
- clean mathematical framework that is used as a foundation for the empirical part
- indication of the fact that DNNs are adapted to the high-dim structures (with certain regularity) that allows to infer them from small portions of data

**Weaknesses:**

- further validation on a more realistic data might be beneficial to strengthen the main points of the paper
- lack of any sort of description of more "general DNN GAHBs"

**Questions:**

N/A

---

> ### Author Response · Authors · 2023-11-20
>
> Thank you for your review.
> - In addition to the LSUN bedroom dataset (in the appendix), we’re continuing to test our method on other datasets, which will be included in the final version of the paper.
> - We acknowledge (in the Discussion) that we don’t (yet) have a mathematically precise definition of GAHBs, which is an interesting open question.

---

### Official Review · Reviewer_SQqg · 2023-11-01

**Soundness:** 3 good
**Presentation:** 4 excellent
**Contribution:** 3 good
**Rating:** 8
**Confidence:** 4

**Summary:**

Diffusion generative models that use denoising DNNs have surpassed previous methods of learning probability models from images. The authors introduce a methodology to evaluate the properties of the trained denoiser and the density from which the data are drawn. They show empirically that diffusion models can converge to a unique continuous density model that is independent of the specific training samples. The convergence exhibits a phase transition between memorization and generalization as training data grows. They demonstrate that two denoising DNNs trained on non-overlapping subsets of a dataset learn nearly the same score function and, thus, the same density, indicating the existence of powerful inductive biases in the DNN architecture and/or training algorithm. The inductive bias of the network appears through the best basis, which is a geometry-adaptive harmonic basis (GAHB) when trained on photographic images. The DNN denoisers achieve near-optimal performance for the Cα class of images. They also investigate the inductive biases that enable this rapid convergence and show that DNN denoisers perform poorly for distributions whose optimal bases are not GAHB. For images drawn from low-dimensional manifolds, DNN denoisers achieve an accurate basis for the subspace and incorporate GAHB vectors in the remaining unconstrained dimensions. The denoiser performs a shrinkage operation on a basis adapted to the underlying image, which reveals oscillating harmonic structures along contours and inhomogeneous image regions. The paper leaves important open questions regarding the formal mathematical definition of this larger class of GAHB bases and how they result from the DNN computational architecture.

**Strengths:**

This is a well-written paper that puts forth a compelling hypothesis.  The hypothesis is presented in a clear and concise manner, and the paper's overall readability is very good. The authors suggest that, instead of attempting to learn a low dimensional structure, DNN denoisers learn a well-regularized geometry-adaptive harmonic basis (GAHB) with small coefficients. Hence, the denoising algorithm performs a shrinkage operation on an image-adapted basis, explaining some of the impressive results that have been observed recently in the literature.

The authors provide a counter-example to argue that if the hypothesis were not true, the DNN denoisers should perform poorly for those images for which GAHB is not the optimal basis. They construct such an example dataset using images drawn from low-dimensional structures and show that the trained DNN denoiser is not perfectly aligned with the manifold, and the bias increases with increasing noise.  Similar results were reported by creating another dataset using shuffled versions of the CelebA dataset, which would also not have GAHB as the optimal basis.

**Weaknesses:**

The paper's results were obtained using a straightforward CNN architecture. However, prior studies have indicated that CNN architectures can effectively learn and utilize harmonic bases (https://arxiv.org/abs/1810.12136). In my opinion, the paper's only weakness lies in not acknowledging this perspective that has been previously established in the literature.

**Questions:**

Would you be able to add some of this line of literature to your discussion?

---

> ### Author Response · Authors · 2023-11-20
>
> Thank you for your review. The paper you mention is indeed interesting, and explains the importance of phase correlations within successive layers of a DNN, and their relevance to image representation.  However, it does not define orthonormal bases and it is not so related to the GAHB bases of the diffusion models, although we do agree that it is worth citing.

---

### Official Review · Reviewer_4Rws · 2023-11-01

**Soundness:** 4 excellent
**Presentation:** 4 excellent
**Contribution:** 4 excellent
**Rating:** 10
**Confidence:** 4

**Summary:**

The authors address recent concerns regarding the memorization of denoising DNNs for generative image modeling by showing that two networks trained on disjoint datasets converge to the same score/density and hence generate similar images. Those experiments are supported by theoretical derivations clearly relating the inductive bias of the denoiser to that of the density. Leveraging recent work, e.g., (Monoho et al.,2020), the authors elucidate the sparse optimal basis, adapted to the geometry of the input image, along which denoising can be understood as a shrinkage operation. Furthermore, the authors validate and connect the results by evaluating the inductive bias on image classes with known optimal bases.

**Strengths:**

- Presents compelling empirical evidence, supported by plausible theoretical derivations, of the inductive bias of denoising DNNs.
- Explains with clear examples how they transition from memorization to generalization (even when trained on disjoint partitions of the datasets).
- Repeats the analysis on 2 datasets.
- Validates on image classes with known optimal bases.

**Weaknesses:**

Nothing stands out, just needs to finalize the text.

**Questions:**

**Presentation:**
- Abstract:
    - Please mention the empirical nature of those findings, while also highlighting the supporting theoretical derivations.
    - It would help to point to the newly highlighted GAHB as a particularly interesting topic for future work, as done on the very last paragraph.
- Section 2:
    - It would help to point to the figures within the main text.
    - Related to this: in many places the authors use figure captions the same way as main text, more so in the appendices. (understandable if that was wrapped up close to the deadline) I strongly recommend to add more supporting text, with pointers to parts in the main text to which each figure is most relevant. That is, to collect those pieces into a coherent narrative that's easy to follow.
- Section 3:
    - S3.1 seems to be largely based on (Mohan et al., 2020). If so, please state this clearly on the onset, or make it clear where the new contributions begin. One point where a citation seemed needed is the notion of projection below Eq.7.

---

> ### Author Response · Authors · 2023-11-20
>
> Thank you for your very positive review, and constructive suggestions, which we will incorporate in the revision.
>
> Regarding S3.1: The use of a bias-free network in equation 6, and the Jacobian analysis of locally-linear behavior, is taken from Mohan et al 2020.  Both citations in the beginning of section 3.1 refer to those contributions. But the rest of section 3 (equation 7 onward) is new and independent of contributions made in Mohan et al 2020.

---

> > ### Comment · Reviewer_4Rws · 2023-11-21
> > **Acknowledgement**
> >
> > Thanks for the response.

---

### Official Review · Reviewer_rH4m · 2023-11-04

**Soundness:** 4 excellent
**Presentation:** 4 excellent
**Contribution:** 4 excellent
**Rating:** 8
**Confidence:** 3

**Summary:**

The authors study the issue of generalization and memorization in diffusion
models. Diffusion models have as a backbone trained denoisers, and it has been
observed that the use of powerful deep networks as denoisers can lead to
situations where diffusion models memorize their training data (rather than
being able to generate novel images from the high-dimensional data
distribution), whereas in other cases the models seem to generalize. The authors
study this for a specific class of deep network denoisers (bias-free CNNs) both
empirically and theoretically. Empirically, they show that training these
denoisers on CelebA and LSUN bedrooms for varying training set sizes witnesses a
clear transition (qualitative and quantitative) between "memorization"
performance of the trained diffusion model when the training set size is small,
and "generalization" performance when it is sufficiently large. The authors
hypothesize that the ability to generalize with relatively few samples from the
high-dimensional distribution is due to inductive biases in the deep network
denoiser, and in particular they make a connection with classical ideas on
denoising from harmonic analysis to posit that DNN denoisers are biased towards
"geometry-adaptive harmonic bases" (eg bandlets/wedgelets).  They test this
hypothesis empirically, demonstrating that these trained denoisers learn
bandlet-like bases in which to perform shrinkage on toy image classes where such
bases are optimal (horizon classes), and moreover that they persist in learning
these types of bases even for classes of signals where they are suboptimal
(image articulation manifolds).

**Strengths:**

- The paper presents detailed (but not overly technical/complex) background on
  diffusion models, to allow a broad audience to appreciate the experimental
  results. (I do wonder here if it might be helpful to present somewhere the
  functional form of the 'optimal' denoiser for the empirical risk, eg as it's
  done in Karras et al 2021, to ground the memorization issue a bit more -- I
  find this concept helpful to have in mind as I'm reading the background of the
  paper.)

- The core question the paper considers -- issues of generalization in diffusion
  models trained to generate samples from high-dimensional data distributions --
  is both important and currently without a completely satisfying
  conceptual/mathematical explanation, despite much concurrent work. The authors
  present a compelling empirical study of this issue under controlled
  conditions, showing (among other things) that it is indeed real, and give a
  plausible theoretical explanation for why it occurs.

- The theory of inductive bias that the authors put forth is nontrivial,
  involving, through equation (6), a specific representational formula for
  piecewise-linear denoisers, and thereby a connection to classical shrinkage
  estimators.

**Weaknesses:**

- It seems that the theoretical framework
  posited in section 3 is specific to the particular denoiser architectures
  being studied in the paper (BF-CNN). For example, PixelNet++-type denoisers
  used in modern diffusion models do not seem to be amenable to a decomposition
  like equation (6), because they involve attention layers and positional
  embeddings (presumably with affine components) to implement different
  conditioning operations. It would be good if the authors could comment on this
  issue, and how they see the theory extending to this modern setting.

**Questions:**

- How robust the main empirical insights are to the
  specific training setup and model architecture being studied in the
  experiments?
  For example, what if one used a modern noise-conditional diffusion model
  instead of training a single unconditioned model on multiple noise scales;
  what if one considered larger-scale training (eg on datasets of
  higher-resolution, photorealistic images, as one considers for modern
  diffusion models); what if one performed generation with an ODE/SDE-type
  sampling procedure, rather than Algorithm 1 used in the paper? It would be
  helpful to know how the authors see Figures 1 and 2 translating into these
  settings, whether they would expect changes, etc.

- Could you clarify behind the thinking behind the claim in the discussion that
  "deep networks are more adapted towards high-dimensional structures [than
  low-dimensional structures]"? Although I think I follow at a high-level --- eg
  in Figure 5, the adaptive basis at the noisy image is not exactly equal to the
  optimal 5-dimensional basis for the low-dimensional tangent space --- it still
  seems to me that there is a bias in what has been learned towards
  low-dimensional structure, because the correct basis for the true tangent
  space seems to be contained in the actual adaptive basis, and the eigenvalues
  of the Jacobian decay at a reasonable rate (making the spectrum overall
  "compressed", if not sparse/low-dimensional). Hence when I try to interrogate
  the claim at a lower level of detail, I cannot exactly 'compile' it.

---

> ### Author Response · Authors · 2023-11-20
>
> Thank you for your in-depth review and questions. In particular, thank you for the suggestion to discuss the behavior of the “memorizing” denoiser, which is optimal for the empirical training loss as opposed to the population validation loss.
>
> 1) Robustness of our results to choices of architecture, dataset characteristics, and the details of the synthesis procedure:
>  - Architecture:
> a) We constrained the architecture to have a zero net bias (Eq 6), which allows the eigen-analysis in section 3.1.  We think this choice is unlikely to affect the results.  In particular, note that Mohan et al 2019 (from which we borrowed the architecture) showed the effective bias of the network tends to zero within the training range of noise levels (in our paper, denoisers are trained on a large range of noise levels).
> b) We have explored the effect of the number of parameters in our architecture (its depth and width). Larger networks with larger images require more training samples to generalize. We will include these results in the appendix for the final revision.
> c) Other major changes in the architecture, however, could potentially result in different inductive biases and  generalization regimes (or even a failure of generalization). An exhaustive exploration over architectures (such as PixelNet++) is an open empirical question which we’ve not addressed. Our method for testing generalization, however, can be used in future to test and detect memorization for any architecture/training scheme. We will make this point clearer in our revision of the discussion.
>
> - Image resolution/size: Increasing the image dimensionality will surely require a larger number of samples to transition to generalization (Figs 1  and 2).  It thus remains open whether state-of-the-art diffusion models have reached the generalization regime or not. We will comment on this in the discussion.
> - Synthesis procedure: we believe that the transition from memorization to generalization is robust to the details of the synthesis algorithm, including the noise-conditional vs. unconditional synthesis. However, this should be verified empirically. We will comment on this in the discussion.
> 2) The sentence "deep networks are more adapted towards high-dimensional structures [than low-dimensional structures]" was indeed cryptic - apologies for that. There is a common belief that deep networks generalize due to the presence of low-dimensional structure in the training data (the “manifold” hypothesis). If networks were indeed inductively biased towards low-dimensional manifolds, we would expect them to be optimal on such distributions (e.g., the disk images, or the single face or sinusoid images in the appendix), and deviate from optimality on higher-dimensional distributions (e.g., the C-alpha family). We observe that the opposite happens, which is indicative of a different inductive bias.

---

> > ### Comment · Reviewer_rH4m · 2023-11-22
> > **thanks**
> >
> > Dear authors,
> >
> > Thanks for your thorough response to my review. I agree that the methodology will be useful and likely impactful for future studies of these issues in broader families of networks and data distributions. I will maintain my score, and congratulate the authors on a great work.

---

### Meta-Review · Area_Chair_uBSA · 2023-12-06

**Metareview:**

This paper investigates the properties of memory and generalization in diffusion models, both experimentally and theoretically. Specifically, the paper identifies the properties of data sets that contribute to the transition phenomena of memory and generalization, and then explains this using a concept that extends the harmonic basis. The results and presentation of the paper are clear and the intent is well communicated to the reader. The reviewers agreed that the paper should be highly commended.

**Justification For Why Not Higher Score:**

N/A

**Justification For Why Not Lower Score:**

We can't find anything to lower the rating of this paper.

---

### Decision · Program_Chairs · 2024-01-16

Accept (oral)